# Preliminary Investigation to Review If a Glycomacropeptide Compared to L-Amino Acid Protein Substitute Alters the Pre- and Postprandial Amino Acid Profile in Children with Phenylketonuria

**DOI:** 10.3390/nu12082443

**Published:** 2020-08-14

**Authors:** Anne Daly, Sharon Evans, Alex Pinto, Richard Jackson, Catherine Ashmore, Júlio César Rocha, Anita MacDonald

**Affiliations:** 1Dietetic Department, Birmingham Children’s Hospital, Steelhouse Lane, Birmingham B4 6NH, UK; evanss21@me.com (S.E.); alex.pinto@nhs.net (A.P.); catherine.ashmore@nhs.net (C.A.); anita.macdonald@nhs.net (A.M.); 2Liverpool Clinical Trials Centre, University of Liverpool, Brownlow Hill, Liverpool L69 3GL, UK; r.j.jackson@liverpool.ac.uk; 3Nutrition and Metabolism, NOVA Medical School, Faculdade de Ciências Médicas, Universidade Nova de Lisboa, 1169-056 Lisboa, Portugal; rochajc@nms.unl.pt; 4Centre for Health Technology and Services Research (CINTESIS), 4200-450 Porto, Portugal

**Keywords:** phenylketonuria, PKU, glycomacropeptide, amino acid, absorption

## Abstract

In Phenylketonuria (PKU), the peptide structure of the protein substitute (PS), casein glycomacropeptide (CGMP), is supplemented with amino acids (CGMP-AA). CGMP may slow the rate of amino acid (AA) absorption compared with traditional phenylalanine-free amino acids (Phe-free AA), which may improve nitrogen utilization, decrease urea production, and alter insulin response. Aim: In children with PKU, to compare pre and postprandial AA concentrations when taking one of three PS’s: Phe-free AA, CGMP-AA 1 or 2. Methods: 43 children (24 boys, 19 girls), median age 9 years (range 5–16 years) were studied; 11 took CGMP-AA1, 18 CGMP-AA2, and 14 Phe-free AA. Early morning fasting pre and 2 h postprandial blood samples were collected for quantitative AA on one occasion. A breakfast with allocated 20 g protein equivalent from PS was given post fasting blood sample. Results: There was a significant increase in postprandial AA for all individual AAs with all three PS. Postprandial AA histidine (*p* < 0.001), leucine (*p* < 0.001), and tyrosine (*p* < 0.001) were higher in CGMP-AA2 than CGMP-AA1, and leucine (*p* < 0.001), threonine (*p* < 0.001), and tyrosine (*p* = 0.003) higher in GCMP-AA2 than Phe-free AA. This was reflective of the AA composition of the three different PS’s. Conclusions: In PKU, the AA composition of CGMP-AA influences 2 h postprandial AA composition, suggesting that a PS derived from CGMP-AA may be absorbed similarly to Phe-free AA, but this requires further investigation.

## 1. Introduction

Protein substitutes are an essential source of synthetic protein in the dietary treatment of classical phenylketonuria (PKU). Protein is the second major constituent in the body, critical for growth and supporting a wide range of metabolic and cellular functions. Amino acids (AA) are engaged in a dynamic process of protein synthesis and degradation. In PKU, it is critical that the AA profile of protein substitutes are carefully developed, with a balance of AAs that meet WHO 2007 [1] minimal AA requirements [2,3]. Furthermore, there is evidence that modification of the large neutral amino acid (LNAA) profile (including tyrosine, leucine, isoleucine, methionine, valine, histidine, threonine and tryptophan) will enhance brain AA concentrations. In mice fed LNAA, an improvement in brain neurophysiology with increased brain serotonin and norepinephrine, and lower phenylalanine concentrations was observed [4]. It is also important that the composition and protein source of protein substitutes help support physiological AA absorption [5].

In PKU, protein substitutes are either manufactured using artificial AA without phenylalanine or based on caseinglycomacropeptide (CGMP), a by-product of the cheese making process [6]. Artificial AAs are generated from plant-based materials converted into sugars, fermented, and purified [7]. In contrast, CGMP modified for use in PKU is a mix of bioactive glycopeptides containing residual phenylalanine with the addition of essential and semi essential artificial AAs (CGMP-AA). Some authors [8,9] describe CGMP as an intact protein. However, proteins are large macromolecules made up of one or more polypeptide chains, whereas CGMP is a 64 macropeptide and therefore classification as an intact protein source is a misrepresentation [10]. Although there is knowledge about the absorption of AAs and intact protein, little is known about the absorption properties of CGMP-AA.

It is well recognized that the kinetic and biochemical properties of natural proteins change depending on the quality of protein, which is determined by its AA pattern, the speed of digestion, absorption and release of AAs into the circulation [11]. Protein metabolism has been extensively studied [12,13]. It is established that whey protein is rapidly absorbed, and the release of some individual AAs (leucine, isoleucine) influences anabolic and hormonal pathways [11,14,15]. In contrast, the appearance of plasma AAs following a meal with casein is slower, with protein synthesis increased and breakdown inhibited [11,16,17]. AAs are directly available for absorption by the small intestine, and so are quickly absorbed, potentially resulting in their transient imbalance, and altering their bioavailability. AA antagonism, the presence of high concentrations of specific AAs, may alter the AA equilibrium, impairing the absorption of other AAs and limiting absorption and metabolism [18]

In PKU, several authors [19,20,21,22] have demonstrated improved utilization of AAs when protein substitutes are taken in divided doses throughout the day. However, this still does not produce a normal physiological response; protein substitutes are usually taken as an addition and not as an integral part of a meal. In an animal study, the metabolic and biological influences of CGMP were compared with Phe-free AA, with CGMP giving a more physiological response decreasing metabolic stress and immunity [23]. In subjects with PKU, van Calcar et al. [24] suggested CGMP improved protein synthesis and nitrogen retention compared to Phe-free AA. However, peptides may be absorbed more rapidly than AA [14,25], thereby questioning the rate of delivery of AAs from CGMP-AA.

In this pilot, parallel study in children with PKU, we aimed to investigate if there were any differences following an overnight fast in the pre and postprandial AA absorption after taking one breakfast dose of Phe-free AA compared with two different CGMP-AA formulations with varying AA compositions.

## 2. Ethical Permission

The South Birmingham Research Ethics committee granted a favorable ethical opinion. The study was registered 13/WM/0435 IRAS (integrated research application system) 129497. Written informed consent was obtained for all subjects from at least one caregiver with parental responsibility and written assent obtained from the subject if appropriate for their age and level of understanding.

## 3. Materials and Methods

### 3.1. Inclusion Criteria

Entry into the study included: children diagnosed with PKU by newborn screening, aged 5–16 years and treated with diet only. Children had to be adherent with diet and protein substitute intake, with 70% of routine blood phenylalanine concentrations within European PKU Guideline target range [26] for six months before study enrolment. Target blood phenylalanine ranges were 120 to 360 µmol/L for children aged 5 up to 12 years and 120 to 600 µmol/L for 12 years and older [26].

### 3.2. Study Design

Pre and postprandial AA absorption was measured on one occasion after six months of taking either Phe-free AA, or one of two CGMP formulations (CGM-AA1 or CGMP-AA2). Children attended the hospital after an overnight fast (minimal fasting time 10 h). CGMP-AA1 had been taken for six months by 11 children as part of a pilot study the results have previously been published [27]. Following the results of the pilot study, a further 18 children were recruited and took CGMP-AA2, which was a modification of CGMP-AA1. Nineteen children remained on Phe-free AA. All children had fasting capillary finger pricks (0.5 mL), for quantitative plasma AAs. Children then took 20 g protein equivalent from Phe-free AA, CGMP-AA1 or CGMP-AA2, followed by a breakfast providing less than one third of their phenylalanine/natural protein allowance (median 2 g natural protein (100 mg phenylalanine), range 1–6). After 120 min post protein substitute and breakfast, a second capillary sample was taken for AAs.

### 3.3. Protein Substitutes (PHE-FREE AA and CGMP1, CGMP2)

The AA profile and nutritional composition of the three different protein substitutes (provided by Vitaflo International) are given in Table 1. All the children in the Phe-free AA group took the same liquid pouch (PKU Cooler 20). For each 20 g protein equivalent, Phe-free AA provided 124 kcal, 9.4 g carbohydrate, and 0.7 g fat, and CGMP-AA1 and CGMP-AA2, 120 kcal, 6.5 g carbohydrate, and 1.5 g fat. CGMP-AA2 had increased amounts of tyrosine, leucine, histidine, and tryptophan, and less methionine, lysine, glycine, and aspartic acid than CGMP-AA1. Except for threonine (higher in CGMP-AA1 and 2), glycine and methionine (higher in CGMP-AA1), and leucine (higher in CGMP-AA2), all the other AAs were slightly but not significantly higher in the Phe-free AA. Glutamine was naturally present in CGMP-AA, but not added to Phe-free AA. The energy content of the three products was similar, although the carbohydrate content was 30% higher in the Phe-free AA, and fat content 53% higher in the two CGMP-AA products, but overall fat intake was low from all three protein substitutes.

The single dose of Phe-free AA, CGMP-AA1, and CGMP-AA2 given in this study provided 20 g protein equivalent. The CGMP-AA1 and CGMP-AA2 also provided an additional 36 mg phenylalanine for each 20 g protein equivalent. The children chose either Phe-free AA or CGMP-AA, depending on their taste preference.

### 3.4. Measurement of Quantitative Plasma Amino Acids

Capillary blood samples were collected into a Sarstedt tube and analyzed by ion exchange HPLC with postcolumn derivatization and spectrophotometric detection (Biochrom, Harvard Bioscience, Holliston, MA, USA). Prior to analysis, separated lithium-heparinized plasma was deproteinized 1:1 with 8% sulphosalicylic acid containing an internal standard, S-2-amino-ethyl-L-cysteine hydrochloride (Sigma, Merck, St. Louis, MO, USA). Quantitative amino acids (QAA) were analyzed except tryptophan and asparagine, which are not reported by our laboratory. Nonproteinogenic AA ornithine, citrulline, and taurine were included in the analysis. The individual pre- and postprandial AAs were quantitated (QAA) and the total AAs, total large neutral amino acids (LNAA), total essential amino acids (EAA), and total branched chain amino acids (BCAA) were calculated from these results. We report total AAs, LNAAs, BCAA, and EAAs, together with individual AAs.

### 3.5. Statistics

Descriptive statistics are reported as medians with associated interquartile ranges. Differences in AAs at baseline and follow-up are assessed using a paired *t*-test. Differences between the three treatment groups are performed using linear regression with differences at follow-up adjusting for baseline covariate values. All analyses are performed in the statistical package R (Version 3.3).

## 4. Results

### 4.1. Subjects

Forty-three (41 European and 2 Asian origin) children with PKU, with a median age of 9 years (range 5–16) were recruited and participated in the study. The number of children in each group was: Phe-free AA, n = 14 (8 boys and 6 girls), CGMP-AA1, n = 11 (5 boys and 6 girls), and CGMP-AA2, n = 18 (11 boys and 7 girls). The median age (range) in the groups were: CGMP-AA1, 8.3 years (6–16), CGMP-AA2, 8.4 years (5–14) and Phe-free AA, 12.9 years (5–15). There was significant difference in age between CGMP-AA2 and the Phe-free AA group (*p* = 0.001). The median phenylalanine concentration for 12 months pre-study (all the children were taking Phe-free-AA) was 288 µmol/L (140–600).

In all three groups, the median daily dose of protein equivalent from protein substitute was 60 g/day (range 40–80 g), and the median amount of prescribed natural protein was 5.5 g/day (range 3–30 g) or 275 mg phenylalanine (range 150–1500 mg). The majority had classical PKU, except two children who were mild according to their untreated blood phenylalanine levels at diagnosis and dietary phenylalanine tolerance.

### 4.2. Quantitative Plasma Amino Acid Results

#### Individual Amino Acids

Significant pre and postprandial differences for most individual AA were observed within each group (Table 2).

Preprandial valine was significantly lower with CGMP-AA2 than CGMP-AA1 (*p* = 0.031) and CGMP-AA2 vs. Phe-free AA (*p* < 0.001). For CGMP-AA1 vs. Phe-free AA, there were no significant preprandial changes.

Postprandial CGMP-AA1 vs. CGMP-AA2: histidine (*p* < 0.001), leucine (*p* < 0.001) and tyrosine (*p* < 0.001) were significantly higher for CGMP-AA2, while methionine (*p* < 0.001) significantly lower compared with CGMP-AA1.

Postprandial CGMP-AA1 vs. Phe-free AA: histidine (*p* = 0.005) and tyrosine (*p* = 0.005) were significantly higher in Phe-free AA, but isoleucine (*p* = 0.008), methionine (*p* < 0.001), threonine (*p* = 0.001) significantly lower. For CGMP-AA2 v Phe-free AA, leucine (*p* < 0.001), threonine (*p* < 0.001), and tyrosine (*p* = 0.003) were all significantly higher in CGMP-AA2.

Changes in the pre and postprandial AA concentrations between the groups appeared to be mainly a reflection of the different amino acid compositions of the three protein substitutes, being most evident between CGMP-AA1 and CGMP-AA2. CGMP-AA2 had higher amounts of histidine, leucine, and tyrosine, and lower methionine and valine compared to CGMP-AA1.

### 4.3. Total Amino Acids, LNAA, BCAA, and EAA

There were similar significant pre and postprandial changes within the groups for total AAs, LNAA, BCAA and EAA (Table 3(a–d), Figure 1a–d). No significant pre or postprandial changes were observed between any of the three groups when comparing total AA, LNAA, BCAA, or EAAs.

## 5. Discussion

This pilot study showed that that the postprandial AA concentrations largely reflected the AA profile of each of the protein substitutes used. CGMP-AA2 contained higher amounts of tyrosine, histidine, and leucine, and lower amounts of methionine and valine compared to CGMP-AA1. These changes were mirrored in the higher postprandial peaks of tyrosine, histidine, and leucine observed between CGMP-AA1 vs. CGMP-AA2. Although there was no postprandial change between the groups for valine, preprandial levels were lower between CGMP-AA1 vs. CGMP-AA2 and CGMP-AA2 vs. Phe-free AA. This is not easily explained physiologically, but may reflect a chance finding, or changes as a result of the competition between the AAs.

It is interesting to speculate on the postprandial changes, as it seems the more AA added to the protein substitute, the higher the AA concentrations were when measured at 120 min. The physiological consequence of these higher AAs is unknown. Postprandial tyrosine was significantly higher in CGMP-AA2 compared with CGMP-AA1 as a direct response of adding extra tyrosine. Norepinephrine is derived from tyrosine and is a principal brain neurotransmitter and so the provision of adequate tyrosine is essential to produce this monaminergic neurotransmitter, which is of clinical significance. Ney et al. [28] measured fasting tyrosine and tryptophan concentrations in subjects taking Phe-free AA compared to CGMP-AA and found their concentrations were 50% higher with Phe-free AA. Gut serotonin levels and microbiome-derived compounds made from tyrosine and tryptophan, although not significantly different, were higher in the CGMP-AA group, suggesting an improved bioavailability of tyrosine and tryptophan [29].

In our study, although individual AAs changed significantly within groups, no significant differences were observed between groups for total AAs, LNAAs, BCAAs, and EAAs. After 120 min, AA concentrations had increased significantly above fasting levels with a 56% increase in CGMP-AA1, 73% increase in CGMP-AA2 and a 42% increase in the Phe-free AA group. The total AA concentration per 20 g protein equivalent for CGMP-AA1 was 21 g, CGMP-AA2, 22 g, and Phe-free AA 24 g. It seems unlikely that the peptide-based CGMP-AA offered any advantage in minimizing the kinetic release of AA, although postprandial bloods were not measured in the first hour post consumption.

This exploratory investigation was a crude assessment to explore if there were any kinetic differences between AAs and a peptide based CGMP-AA with a different AA profile. In a crossover study in eleven adults with PKU, MacLeod et al. [30] measured postprandial AAs after 180 min following a breakfast with Phe-free AA or CGMP-AA. CGMP-AA was consumed as GMP foods rather than drinks. In the CGMP-AA group, postprandial threonine and isoleucine were significantly higher, and total AA concentrations just reached a significant difference compared to the Phe-free AA group. The authors suggested that based on the higher concentrations of insulin and total plasma AAs in the GMP group, CGMP-AA had an improved AA absorption profile compared to Phe-free AA. Although the preprandial breakfast was isocaloric, the AA composition of both products was not stated and the protein substitute in the form of a food versus a liquid may alter the absorption of AAs. Similarly, a non-physiological response causing a rapid rise in insulin concentrations may not be ideal. What remains unknown both in this and our own study is at what point maximum and nadir concentrations for AAs were reached, and neither study measured concentrations over 240 min or used a whole protein source as a comparison from which maximum and minimum AA concentrations could be compared.

Ahring et al. [31] compared two groups of protein substitutes: group 1, CGMP only versus Phe-free AA (different protein sources but the same AA composition) and group 2, CGMP-AA (CGMP with added AAs) versus Phe-free AA, (different protein sources and the same AA composition). The AA profile was different between group 1 and 2. Measurements were made over 240 min. They reported no differences in the absorption of total AAs between the two groups, suggesting that CGMP made no impact on the absorption rate of AAs. However, some individual AAs changed significantly between the groups. For example, the tyrosine amounts in the different product consumed were markedly different; group 1, 0.05 g and group 2, 10.81 g. At 30 min post ingestion, plasma tyrosine concentrations in group 2 were double those in group 1, reflecting changes in the AA profile of the protein substitute rather than the source of protein.

Gropper et al. [32] and others [33,34,35] have demonstrated that the type and quality of protein influences kinetic absorption. Healthy volunteers ingested one of three protein sources: AAs, a mixture of 75% AA with 25% natural protein, or whole protein. After 150 min, the only AA profile significantly higher than baseline was the group ingesting whole protein. In the two other groups, peak AA concentrations were reached before this time point. These studies showed that peak AA concentrations from a Phe-free AA or Phe-free AA combined with an intact protein source were more rapidly absorbed compared with an intact protein source only.

Both the time of arrival and pattern of AAs in the systemic circulation are important for effective protein synthesis. For this to occur efficiently all essential AAs must be presented to the tissues in appropriate amounts simultaneously. In PKU, the delivery of AAs to tissues is accelerated compared to a diet based on mixed proteins [12]. Glutamine is the most abundant free AA in the body, with a wide range of diverse molecular actions [36]. Its primary source is skeletal muscle. Both BCAA and lysine, by different mechanisms, act as precursors for glutamine synthesis, and leucine can stimulate the release of glutamine and alanine from muscles. Threonine in high concentrations can decrease glutamine formation [37]. The importance of understanding the delivery of AAs from protein substitutes and their effect on molecular pathways is crucial to long-term health outcomes for patients reliant on protein substitutes for their main source of nitrogen.

There are limitations to our findings. This was a pilot study with the aim to explore AA absorption from protein substitutes with different AA compositions. The liquid Phe-free AA preparation contained 30% more carbohydrate and 50% less fat than the CGMP-AA preparations. Breakfast provided similar food choices and the protein content was controlled, providing no more than 30% of their natural/phenylalanine daily allowance; however, we did not standardize the breakfast for all subjects, nor did we measure the AA concentrations every 30 min or over the course of 240 min as recommended by others [11,37], thereby missing the peak and baseline values. Similarly, we did not collect any other supporting data such as insulin and glucose concentrations to review the effect of insulinotropic AAs between the different protein sources. Our AA analysis did not measure tryptophan. We did not compare children with PKU with a control group taking a standard breakfast only without protein substitute.

## 6. Conclusions

In conclusion, the delivery, timing, and ratios of AAs are essential to maximize nitrogen utilization and biochemical functions. This pilot investigation compared three protein substitutes with different AA compositions and two different protein sources. It appeared that the AA composition rather than protein source was more important in determining postprandial plasma AAs. Further detailed work is needed to understand the kinetic and functional roles of protein substitute based on different protein sources and their metabolic impact

## Figures and Tables

**Figure 1 nutrients-12-02443-f001:**
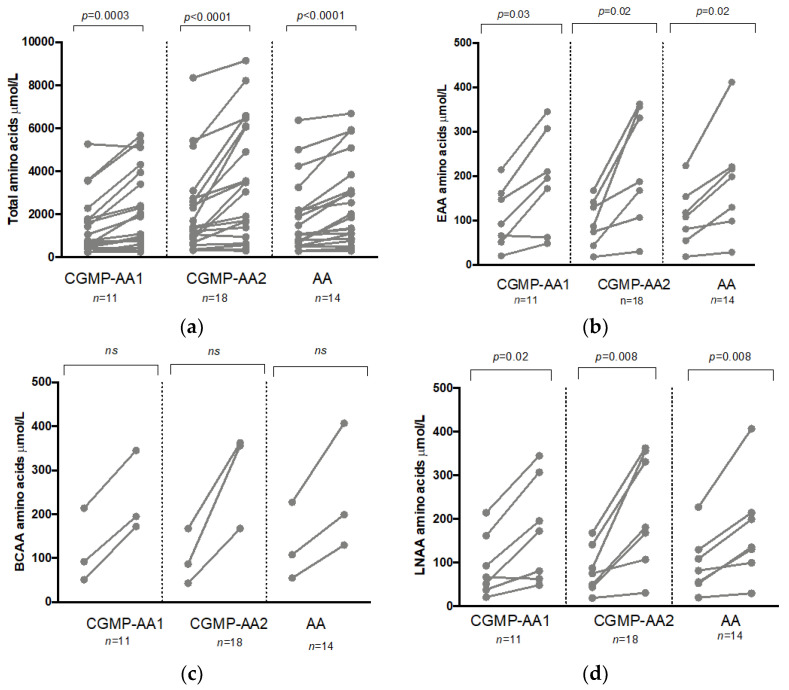
(**a**) Total median pre and postprandial amino acid concentrations for total amino acids (*n* = 17) for CGMP-AA1, CGMP-AA2, and PHE-FREE AA protein substitutes. (**b**). Total median pre and postprandial amino acid concentrations for EAA (*n* = 7) for CGMP-AA1, CGMP-AA 2, and PHE-FREE AA protein substitutes. (**c**) Total median pre and postprandial amino acid concentrations for BCAA (*n* = 3) for CGMP-AA1, CGMP-AA2, and PHE-FREE AA protein substitutes. (**d**) Total median pre and postprandial amino acid concentrations for LNAA (*n* = 8) for CGMP-AA1, CGMP-AA2, and PHE-FREE AA protein substitutes.

**Table 1 nutrients-12-02443-t001:** Nutritional composition of CGMP-AA 1, CGMP-AA2, and PHE-FREE AA protein substitutes.

Protein Substitute		CGMP-AA1	CGMP-AA2	PHE-FREE AA
Nutrients	Units	Per 20 g PE Sachet	Per 20 g PE Sachet	Per 20 g PE Pouch
Calories	Kcal	120	120	124
Protein equivalent	g	20	20	20
Total Carbohydrate	g	6.5	6.5	9.4
Sugars	g	2.2	2.2	7.8
Total Fat	g	1.5	1.5	0.7
Docosahexaenoic Acid	mg	84	84	134
Arachidonic Acid	mg	-	-	-
Fiber	g	0.1	0.1	-
Comprehensive amino acid profile				
L-amino acids		**CGMP-AA1**	**CGMP-AA2**	**PHE-FREE AA**
		**20 g PE**	**20 g PE**	**20 g PE**
L-Alanine	g	0.76	0.83	0.92
L-Arginine	g	1.00	0.96	1.5
L-Aspartic Acid	g	2.04	1.31	2.37
L-Cystine	g	0.01	0.24	0.61
L-Glutamine	g	2.49	2.70	-
Glycine	g	2.77	1.20	2.35
L-Histidine	g	0.42	0.70	0.92
L-Isoleucine	g	1.37	1.35	1.62
L-Leucine	g	1.30	3.00	2.54
L-Lysine	g	1.07	0.80	1.67
L-Methionine	g	0.54	0.28	0.45
L-Phenylalanine	g	0.03	0.03	-
L-Proline	g	1.51	1.52	1.69
L-Serine	g	0.98	0.96	1.04
L-Threonine	g	2.17	2.20	1.62
L-Tryptophan	g	0.17	0.40	0.5
L-Tyrosine	g	1.01	2.24	2.38
Taurine	g	-	-	0.04
L-Valine	g	1.13	1.09	1.86

CGMP-AA 1/CGMP-AA 2: casein glycomacropeptide formula 1 and 2; PHE-FREE AA: Phenylalanine-free L-amino acid (PKU Cooler 20, Vitaflo International).

**Table 2 nutrients-12-02443-t002:** Median (range) fasting pre and postprandial individual amino acids for CGPM-AA1, CGMP-AA2, and PHE-FREE AA protein substitutes.

Median L-Amino Acids μmol/L (range)	CGMP-AA1 (*n* = 11)		CGMP-AA 2 (*n* = 18)		PHE-FREE AA (*n* = 14)		*p* Value
	Pre- Prandial	Post- Prandial	Pre- Prandial	Post- Prandial	Pre- Prandial	Post- Prandial	
Alanine	320 ^ (171–425)	482 ^^,^** (266–685)	277 ^^^,§^ (182–566)	457 ^^ (304–701)	356 ^^^^,§^ (153–463)	412 ^^^^,^** (319–612)	^ 0.007, ^^ <0.0001, ^^^ <0.002 ^§^ 0.012, ** 0.036
Arginine	44 ^ (27–56)	77 ^ (28–138)	38 ^^ (14–56)	94 ^^ (23–130)	38 ^^^ (16–57)	74 ^^^ (33–168)	^ 0.003, ^^ <0.0001, ^^^ 0.0001
Aspartic acid	21 ^T^ (12–35)	19 (12–36)	17 ^T^ (11–28)	15 (10–31)	21 (6–31)	18 (11–42)	^T^ 0.03
Citrulline	39 ^T^ (30–51)	39 (28–55)	29 ^T^ (23–46)	33 (24–49)	37 (16–48)	35 (17–47)	^T^ 0.02
Cystine	36 ^^,T^ (18–45)	28 ^ (12–44)	16 ^^^,T,§^ (4–35)	21 ^^^,§§^ (5–35)	30 ^§^ (19–57)	31 ^§§^ (20–171)	^ 0.02, ^^ 0.001 ^T^ 0.003, ^§^ <0.001, ^§§^ <0.001
Glutamine	495 ^ (406–542)	476 ^^,TT^ (258–578)	472 ^^ (350–606)	513 ^^^,TT^ (396–675)	461 (339–583)	455 (345–613)	^ 0.001, ^^ 0.001 ^TT^ 0.035
Glutamic acid	62 ^ (40–79)	71 ^ (45–104)	63 ^^ (31–133)	70 ^^ (41–155)	75 (22–110)	78 (33–111)	^ 0.04, ^^ 0.02
Glycine	323 ^ (227–429)	476 ^^,TT,^** (291–767)	306 ^^ (195–446)	342 ^^^,TT^ (227–526)	303 ^^^ (189–380)	368 ^^^^,^** (199–642)	^ 0.003, ^^ <0.0001, ^^^ 0.004 ^TT^ <0.001, ** 0.001
Histidine	66 (56–83)	62 ^TT,^** (39–83)	75 ^^ (62–119)	107 ^^^,TT^ (70–151)	81 ^^^ (46–99)	99 ^^^^,^** (53–181)	^^ <0.0001, ^^^ 0.003 ^TT^ <0.001, ** 0.005
Isoleucine	51^ (39–71)	172 ^^,^** (97–270)	43 ^^ (34–62)	168 ^^ (96–230)	55 ^^^ (30–72)	130 ^^^^,^** (68–311)	^ 0.001, ^^ <0.0001, ^^^ 0.0001 ** 0.008
Leucine	92 ^ (73–116)	195 ^^,TT^ (99–296)	87 ^^ (72–130)	356 ^^^,TT,§§^ (227–440)	108 ^^^ (66–127)	199 ^^^^,§§^ (108–474)	^ 0.001, ^^ <0.0001, ^^^ 0.0001 ^TT^ <0.001, ^§§^ <0.001
Lysine	147 ^ (102–179)	210 ^ (128–314)	130 ^^^,§^ (94–172)	188 ^^ (96–332)	155 ^^^^,§^ (109–193)	216 ^^^ (148–385)	^ 0.002, ^^ <0.0001, ^^^ 0.0001 ^§^ 0.04
Methionine	20 ^ (13–23)	48 ^^,TT,^** (28–82)	18 ^^ (15–27)	30 ^^^,TT^ (15–42)	19 ^^^ (14–27)	29 ^^^^,^** (19–52)	^ 0.001, ^^ <0.0001, ^^^ 0.002 ^TT^ <0.001, ** <0.001
Ornithine	66 ^ (53–114)	87 ^ (55–175)	72 ^^ (37–96)	91 ^^ (69–136)	83 ^^^ (37–99)	99 ^^^ (51–141)	^ 0.004, ^^ 0.0004, ^^^ 0.004
Proline	116 ^ (93–183)	282 ^ (174–522)	106 ^^ (80–284)	275 ^^ (198–530)	141 ^^^ (83–322)	258 ^^^ (161–447)	^0.002, ^^ <0.0001, ^^^ 0.0001
Serine	156 ^ (123–227)	202 ^ (131–321)	146 ^^ (119–196)	193 ^^ (119–251)	155 (71–228)	169 (132–325)	^ 0.008, ^^ <0.0001
Taurine	54 (35–73)	53 (36–326)	55 (26–92)	47 (33–74)	56 (39–98)	56 (47–100)	
Threonine	161 ^ (90–241)	307 ^^,^** (200–614)	141 ^^ (80–215)	331 ^^^,§§^ (240–440)	129 ^^^ (67–193)	214 ^^^^,^**^,§§^ (122–358)	^ 0.002, ^^ <0.0001, ^^^ 0.0001 ^**^ 0.001, ^§§^ <0.001
Tyrosine	37 ^ (29–68)	80 ^^,TT,^** (34–149)	49 ^^ (36–84)	181 ^^^,TT,§§^ (126–327)	47 ^^^ (31–73)	136 ^^^^,^**^,§§^ (52–206)	^ 0.002, ^^ <0.0001, ^^^ 0.0001 ^TT^ <0.001,** 0.005, ^§§^ 0.003
Valine	214 ^^,T^ (157–278)	345 ^ (206–580)	168 ^^^,§,T^ (131–243)	363 ^^ (248–538)	227 ^^^^,§^ (136–345)	407 ^^^ (240–628)	^ 0.002, ^^ <0.0001,^^^ 0.0001 ^T^ 0.031, ^§^ <0.001

Key: ^, ^^, ^^^ significant differences within CGMP1,2 and Phe-free AA. Preprandial changes: T = CGMP1 v2, ^§^ = CGMP2 vPhe-free AA. Postprandial changes TT = CGMP1v2, ** = GGMP1 vs. Phe-free AA, ^§§^ = CGMP2 vs. Phe free AA. *p* = significant value

**Table 3 nutrients-12-02443-t003:** (**a**) Median pre and postprandial total amino acids results for CGMP-AA1, CGMP-AA2, and PHE-FREE AA protein substitutes. (**b**) Median pre and postprandial EAA for CGMP-AA1, CGMP-AA2, and PHE-FREE AA protein substitutes. (**c**) Median pre and postprandial BCAA for CGMP-AA1, CGMP-AA2, and PHE-FREE AA protein substitutes. (**d**) Median pre and postprandial LNAA for CGMP-AA1, CGMP-AA2, and PHE-FREE AA protein substitutes.

			(**a**)				
	**CGMP-AA1 *n*= 11**		**CGMP-AA2 *n* = 18**		**PHE-FREE AA *n* = 14**		***p* Value**
Total AA μmol/L (range)	**Pre-** **Prandial**	**Post-** **Prandial**	**PRE-** **PRANDIAL**	**Pre-** **Prandial**	**Pre-** **Prandial**	**Pre-** **Prandial**	
	753 * (219–5257)	1473 * (237–5659)	1375 ** (317–8344)	3249 *^*^ (291–9139)	1067 *** (336–8513)	1922 *** (396–9064)	* 0.0003, ** <0.0001, *** <0.0001
**Amino acids:** alanine, arginine, aspartic acid, cystine, glutamine, glutamic acid, glycine, histidine, isoleucine, leucine, lysine, methionine, proline, serine, threonine, tyrosine, valine (excluding phenylalanine, tryptophan, citrulline)							
(**b**)							
EAA μmol/L (range)	**Pre-** **Prandial**	**Pre-** **Prandial**	**Pre-** **Prandial**	**Pre-** **Prandial**	**Pre-** **Prandial**	**Pre-** **Prandial**	
	92 * (20–214)	195 * (48–345)	87 ** (18–168)	188 ** (30–363)	108 *** (17–223)	199 *** (30–415)	* 0.03 ** 0.02 *** 0.02
**Amino acids:** histidine, isoleucine, leucine, lysine, methionine, threonine, valine (excluding phenylalanine, tryptophan)							
(**c**)							
BCAA μmol/L (range)	**Pre-** **Prandial**	**Pre-** **Prandial**	**Pre-** **Prandial**	**Pre-** **Prandial**	**Pre-** **Prandial**	**Pre-** **Prandial**	
	92 (51–214)	195 (172–345)	87 (43–168)	356 (168–363)	98 (46–223)	214 (131–415)	ns
**Amino acids:** isoleucine, leucine, valine							
(**d**)							
LNAA μmol/L (range)	**Pre-** **Prandial**	**Pre-** **Prandial**	**Pre-** **Prandial**	**Pre-** **Prandial**	**Pre-** **Prandial**	**Pre-** **Prandial**	
	66 * (20–214)	172 * (48–345)	75 ** (18–167)	180 ** (30–363)	67 *** (17–223)	138 *** (30–415)	* 0.03, ** 0.02, *** 0.02
**Amino acids:** histidine, isoleucine, leucine, methionine, threonine, tyrosine, valine							

CGMP-AA 1, CGMP-AA 2: caseinglycomacropeptide formula 1 and 2, PHE-FREE AA: Phenylalanine-free L-amino acids._*, **, ***_
*p* = significant value, ns not significant.

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
