# Peer review of "Preliminary Investigation to Review If a Glycomacropeptide Compared to L-Amino Acid Protein Substitute Alters the Pre- and Postprandial Amino Acid Profile in Children with Phenylketonuria"

_nutrients, 2020, doi:10.3390/nu12082443_

Round 1
Reviewer 1 Report
I find this paper well written and well representing the research done. Suggestions include:
Line 41: place (PS) after what I believe to be this first use of protein substitute in the paper body
Line 52: describing protein substitute formulas based on free amino acids as " artificial AA" would be better if clarified to a more descriptive "free AA" or "single AA", since these AAs are obtained from plant sources and not manufactured chemically.
Line 98: instead of abbreviating protein substitutes composed of free amino acids as only "AA" it would be more clear if a term such as PS composed of free AA, or similar was used. This eliminates confusion between product and blood AAs.
Line 250: Gropper's name is repeated at the beginning of the sentence.
General comments: I find this preliminary research interesting and well thought out. I suggest that in further study it would be important to provide a standard breakfast to make blood amino acid results less confused by an unknown amount and composition of amino acids taken in food.
Author Response
Reviewer 1
I find this paper well written and well representing the research done. Suggestions include:
Line 41: place (PS) after what I believe to be this first use of protein substitute in the paper body.
Thank you. We have changed PS to protein substitute throughout the main body of the text. We have retained the abbreviation PS in the abstract and placed (PS) after the first use of protein substitute.
Line 52: describing protein substitute formulas based on free amino acids as " artificial AA" would be better if clarified to a more descriptive "free AA" or "single AA", since these AAs are obtained from plant sources and not manufactured chemically.
Thank you. We have changed the word manufactured to artificial AA.
Line 98: instead of abbreviating protein substitutes composed of free amino acids as only "AA" it would be more clear if a term such as PS composed of free AA, or similar was used. This eliminates confusion between product and blood AAs.
Thank you. We have changed AA to Phe-free AA throughout the document to add clarity.
Line 250: Gropper's name is repeated at the beginning of the sentence.
Thank you. This sentence has been changed and likewise all other references that appeared with two names.
General comments: I find this preliminary research interesting and well thought out. I suggest that in further study it would be important to provide a standard breakfast to make blood amino acid results less confused by an unknown amount and composition of amino acids taken in food.
Thank you. We agree completely. This was a preliminary work and we did not expect to find the results we did. We will be starting a controlled kinetic study shortly.
Reviewer 2 Report
General comments
The authors present a “preliminary” investigation comparing glycomacropeptide products to an amino acid product and their effect on amino acid levels in the blood. The study is not rigorously controlled, so cannot provide definitive answers to many of the scientific questions one would like answered with this kind of study. However, I feel the data obtained is useful information for the scientific community, the authors freely acknowledge the limitations of the study, and the discussion is thoughtful.
Specific comments
Abstract
Line 25-26: “slow the rate of amino acid (AA) absorption” – suggest you mention this is compared to standard amino acid formulae (this becomes implicit in Line 28, but still worth stating the issues clearly from the start)
Line 28: “…with AA protein substitute (PS)” – My first reading of this was that you were calling the third group (the standard amino acid formula group) as “AA protein substitute (PS)”. Perhaps you could mention protein substitutes in the first sentence and define PS there, which would also address the previous comment.
Materials and Methods
Line 109-110: “carbohydrate…30% higher in the CGMP-AA “ – it actually appears to be higher in the AA product (9.4g vs 6.5g). Correctly stated in the discussion (Line 269)
Table 1 / Line 112-113: AA supplement has ½ the fat – not commented on, and this might affect absorption rate
Table 1 / Line 113-116: CGMP is a mixture of peptides and free amino acids (as per Line 56). If the central tenet underlying the hypothesis for the study is that peptides are absorbed differently than free amino acids, if would be useful to have those distinguished when the AA composition of CGMP-1 and 2 are detailed in Table 1. Conversely, having the values for 1g PE is not that valuable when the 20g levels are there, so perhaps the table could use that space to indicate how much is from the peptide (or from AA supplementation) and the total for each AA.
Table 1 – the micronutrient content is irrelevant for this study
Results
Line 151: natural protein range 3-30g – If each child received 1/3 of their daily protein/phenylalanine as the remainder of the breakfast meal (line 135), this means that there was huge variation (10 fold) in the natural protein each child consumed with the 20g protein-equivalent test dose. I think this study design complicates the interpretation compared to using a fixed natural protein load (e.g. 1.5 g). This is briefly mentioned in the discussion (Line 271), but what effect this might have had on the data is not explored.
Table 2 – the reporting of the significant level comparisons looks daunting. (Suggestion) Perhaps a consistent indicator for each possible comparison (e.g. pre-post for AA is always ^^^, Pre-pre CGMP-AA2 to AA is always *, etc) and consistent placement (i.e. there are 2 lines available for each AA, line 1 could always be the pre-post comparisons and line 2 the between group comparisons) would be easier for the reader to scan.
Table 3 – similar comments to Table 2 (also, could use the same symbols to mean the same thing between tables)
Figure 1 – I do not think this figure provides additional information that is not already in the tables. On first glance, one expects that the figures are actually showing the individual subject levels, which would be new (and perhaps useful) information. Suggest delete the figure or change to individual subject pre-post levels for total AA, EAA, BCAA, LNAA.
Figure 1 – the micromole symbol in the y-axis title is not displaying correctly in the version I downloaded. This might be specific to my computer/software.
Line 233 – It is not obvious that having higher insulin peaks is an improvement
Author Response
Dear reviewer,
Please check the attachment. Thank you very much for your review.
